# Displacement of Gray Matter and Incidence of Seizures in Patients with Cerebral Cavernous Malformations

**DOI:** 10.3390/biomedicines9121872

**Published:** 2021-12-10

**Authors:** Chi-Jen Chou, Cheng-Chia Lee, Ching-Jen Chen, Huai-Che Yang, Syu-Jyun Peng

**Affiliations:** 1Division of Neurosurgery, Department of Surgery, Kaohsiung Veterans General Hospital, Kaohsiung 813414, Taiwan; ck911046@gmail.com; 2Department of Neurosurgery, Neurological Institute, Taipei Veterans General Hospital, Taipei 112201, Taiwan; yfnaughty@gmail.com (C.-C.L.); wade012@gmail.com (H.-C.Y.); 3School of Medicine, National Yang Ming Chiao Tung University College of Medicine, Taipei 11221, Taiwan; 4Brain Research Center, National Yang Ming Chiao Tung University, Taipei 11221, Taiwan; 5Department of Neurological Surgery, University of Virginia Health System, Charlottesville, VA 22903, USA; chenjared@gmail.com; 6Professional Master Program in Artificial Intelligence in Medicine, College of Medicine, Taipei Medical University, Taipei 10675, Taiwan

**Keywords:** cerebral cavernous malformation, seizure, displaced gray matter, biomedical imaging analysis

## Abstract

Seizures are the most common presentation in patients with cerebral cavernous malformations (CCMs). Based on the hypothesis that the volume or proportion of gray matter (GM) displaced by CCMs is associated with the risk of seizure, we developed an algorithm by which to quantify the volume and proportion of displaced GM and the risk of seizure. Image analysis was conducted on 111 patients with solitary CCMs (divided into seizure and nonseizure groups) from our gamma knife radiosurgery (GKRS) database from February 2005 and March 2020. The CCM algorithm proved effective in quantifying the GM and CCM using T1WI MRI images. In the seizure group, 11 of the 12 patients exhibited seizures at the initial presentation, and all CCMs in the seizure group were supratentorial. The location of the limbic lobe within the CCM was significantly associated with the risk of seizure (OR = 19.6, *p* = 0.02). The risk of seizure increased when the proportion of GM displaced by the CCM exceeded 31%. It was also strongly correlated with the volume of displaced GM. The volume and proportion of displaced GM were both positively correlated with the risk of seizure presentation/development and thus could be used to guide seizure prophylaxis in CCM patients.

## 1. Introduction

Seizures are the most common manifestation of cerebral cavernous malformation (CCM) [1]. In a prospective follow-up study, the annual incidence of seizure in patients with CCMs was estimated at 1.5% to 2.4% [2]. Cortical involvement and CCM location (supratentorial, archicortical, and mesiotemporal) have been identified as risk factors for seizures [2,3]. Researchers have proposed hemosiderin deposition related to hemorrhage as the mechanism underlying CCM epileptogenicity [4]. Researchers have also reported that free radicals and lipid peroxides associated with the breakdown of hemoglobin induce excitotoxicity in adjacent neurons and the proliferation of the glial tissue by interrupting receptor activity, calcium hemostasis, and neurotransmitters [5].

Researchers have identified cortical involvement as the most important risk factor for seizure development in cases of CCM. Specifically, mesiotemporal lobe involvement has been associated with an elevated risk of seizure development [2,6]. Nonetheless, researchers have yet to elucidate the relationship between the volume or proportion of cortical involvement and the risk of seizure risk. In the current study, we sought to overcome the difficulties in the manual analysis of cortical involvement by developing an algorithm by which to quantify the volume and proportion of displaced gray matter as well as the corresponding correlations with the risk of seizure.

## 2. Materials and Methods

### 2.1. Patient Population

Between February 2005 and March 2020, 149 patients with CCMs underwent gamma knife radiosurgery (GKRS) at Taipei Veterans General Hospital (TVGH); 29 patients were excluded due to multiple CCMs. Another 9 patients were excluded due to a lack of complete imaging records for analysis, which left 111 patients for analysis. Treatment imaging data retrieved from the GKRS plans included T1-weighted images (T1WI) and T2-weighted images (T2WI). We reviewed all of the patient medical records in terms of initial symptoms, medications, and seizure presentation. Patients who had at least one seizure episode or were taking at least one antiepileptic drug were assigned to the “seizure” group. Patients without a history of seizure and those not taking antiepileptic drugs were assigned to the “nonseizure” group.

### 2.2. MRI Protocol

All magnetic resonance imaging (MRI) scans of CCM patients were obtained using a Signa HDxt 1.5 T (GE Healthcare Milwaukee, WI) scanner with an eight-channel phase-array neurovascular coil. The parameters used to acquire the T1WI sequences were as follows: repetition time (TR) = 400–500 ms, echo time (TE) = 8–9 ms, field of view (FOV) = 260 mm, number of excitations (NEX) = 2, slice thickness = 3 mm, and spacing between slices = 3 mm. The parameters used to acquire the T2WI sequences were as follows: TR = 3466–4050 ms, TE = 100–112 ms, FOV = 260 mm, NEX = 2, slice thickness = 3 mm, and spacing between slices = 3 mm. We created a density template of gray matter (GM) in normal control by which to compare the volume and proportion of GM displaced by the CCM. Specifically, we enrolled 22 normal subjects who underwent MRI analysis at TVGH using a 3T scanner with a 20-channel head coil (Discovery MR750, GE Healthcare) and a 3D T1 fast spoiled gradient echo sequence (FSPGR) using the following imaging parameters: TR = 9.384 ms, TE = 4.036 ms, flip angle = 12°, and voxel size = 1.0 × 1.0 × 1.0 mm^3^. The age range of the 22 normal control subjects was 20 to 30 years old, with a mean age of 25.3 years and a standard deviation of 4.4 years.

### 2.3. Quantitative CCM Evaluation

We developed an automated CCM algorithm capable of quantifying the GM and CCM in T1WI MRI images from GKRS plans with the aim of estimating the volume and proportion of GM displaced by the CCM. The algorithm was run on a PC with an Intel Core™ i7-9750H CPU@ 2.60 GHz 2.59 GHz with 32 GB RAM. Analysis was conducted primarily in the MATLAB R2021a environment (MathWorks, Inc., Natick, MA, USA) using the statistical parametric mapping program, SPM12 (Functional Imaging Laboratory, Institute of Neurology, University College London, London, UK). Three-dimensional FSPGR images from normal control subjects and T1WI images of CCM patients were respectively subjected to image processing in two distinct stages.

#### 2.3.1. Stage 1–Preprocessing Control MPRAGE Images

We first processed 3D FSPGR images of normal control subjects to establish a GM density template (in the Montreal Neurological Institute space) specific to local subjects. Stage 1 was implemented in four steps, as follows:Step C1. Image format conversion

Three-dimensional FSPGR images of normal control subjects were used to segment brain tissue for use in constructing the GM density template. The original digital imaging and communications in medicine (DICOM) file format of 3D FSPGR images was converted into the 3D NIfTI-1 file format to facilitate subsequent image processing using SPM12.

Step C2. Resetting image orientation

The origin of the 3D FSPGR image was shifted to align with the anterior commissure of the individual brain space. Our aim in this step was to improve 3D FSPGR image segmentation performance and facilitate normalization of the GM density map to the standard brain space in the following step.

Step C3. Spatial normalization to the Montreal Neurological Institute (MNI) space

Three-dimensional FSPGR images were segmented to obtain GM density maps and spatially aligned to the Montreal Neurological Institute (MNI) space using the diffeomorphic anatomical registration and the exponentiated lie algebra (DARTEL) module in SPM12 [7]. The dimensions of the 3D FSPGR images and GM density maps were downsized from the original 512 × 512 × 160 voxels to 181 × 217 × 181 voxels.

Step C4. Establishing a customized GM density template in the MNI space

Normalized GM density maps from the 22 normal control subjects were averaged voxel-wise across subjects to create a customized GM density template in the MNI space.

#### 2.3.2. Stage 2—Preprocessing of T1WI Images from Patients

As shown in Figure 1, our objective in Stage 2 was to quantify the volume as well as the proportion of GM displaced by the CCM and then calculate the proportions of the eight lobes (within the CCM) in the MNI space. Stage 2 was implemented in six steps, as follows:Step P1. Conversion of DICOM to NIfTI format

This step was identical to Step C1 above, albeit using patient T1WI as the input.

Step P2. Resetting the T1WI orientation

This step was identical to Step C2 above, albeit using patient T1WI images.

Step P3. Manual delineation of the CCM

Manual delineation of the CCM was performed by an experienced neurosurgeon (Lee, CC) using T2WI images (from the GKRS planning system) to produce CCM image labels.

Step P4. Normalizing T1WI and CCM image labels to the MNI space

Processed T1WI images and CCM image labels were spatially aligned with the SPM T1 template of the MNI space using the normalization module in SPM12. This resulted in voxel-to-voxel coregistration between patient-normalized CCM image labels and the customized GM density template.

Step P5. Quantifying the displacement of GM by CCM

The volume and proportion of GM displaced by the CCM region were quantified via statistical analysis of normalized CCM image labels using the customized GM density template as a reference. In the GM density template, regions with density of >0.5 were designated as gray matter (GM), whereas regions with density of <0.5 were designated as white matter (WM) or cerebrospinal fluid (CSF). The volume of displaced GM was estimated as follows:
(1)$$V_{displace\;GM}(\mathrm{ml})=Nd_{ccm}\times Vs/1000$$
where $V_{displace\;GM}(\mathrm{ml})$ indicates the volume of displaced GM, *Nd_ccm_* indicates the number of voxels for GM (where density >0.5) within the region of the CCM, and
*Vs* refers to the size of the voxel (mm^3^).

The proportion of displaced GM was estimated as follows:(2)$$P_{\mathrm{displace}\;GM}\;(\%)=\sum {D_{k}}_{\;}/\;Nccm$$
where $P_{displace\;GM}\left(\%\right)$ indicates the proportion of displaced GM, $D_{k}$ indicates the density of voxel *k*, ∑*D_k_* refers to the sum density of GM within the region of the CCM, and $N_{ccm}$ refers to the total number of voxels within the region of the CCM.

Step P6. Quantifying lobe proportions within the CCM

The TD atlas (WFU PickAtlas 3.0.5) was used to quantify the proportion of each lobe within the CCM [8,9]. The eight lobes included the cerebellum (merged cerebellum anterior lobe and cerebellum posterior lobe), frontal lobe (merged frontal lobe and frontal-temporal space), limbic lobe, brain stem (merged medulla, midbrain, and pons), occipital lobe, parietal lobe, temporal lobe, and sublobar for quantitation analysis. The proportions of the eight lobes within the CCM were calculated in the MNI space.

### 2.4. Statistical Analysis

Descriptive statistics are presented as the count and mean ± standard deviation. Wilcoxon rank sum tests, Fisher’s exact test, and logistic regression were used for group comparisons (seizure and nonseizure patients) of continuous or categorical variables. The level of statistical significance was set at *p* < 0.05. The statistical toolbox of MATLAB R2021a was used for data analysis. The power of the model in discriminating between seizure and nonseizure groups was assessed in terms of sensitivity, specificity, positive predictive value (PPV), negative predictive value (NPV), F1-score, diagnostic odds ratio (DOR), Matthews correlation coefficient (MCC), kappa coefficient and receiver operating characteristic (ROC) curve, area under the curve (AUC), and calibration plots. The Hosmer–Lemeshow test was also used to test the goodness of fit for logistic regression models. We assessed the predicted PPV based on the prevalence of seizures in the CCM population. Bayesian rules were used to estimate the performance of the proposed method in a practical situation. Predictions of PPV in the CCM population were estimated as follows:(3)predictedPPV=(Se×P)/(Se×P+(1−Sp)×(1−P)),
where *P* refers to true prevalence of seizure in the CCM population, and *Se* and *Sp*, respectively, indicate the sensitivity and specificity when applied to our samples. All statistical analysis was conducted using the SAS Suite, version 9.4 (SAS Institute, Cary, NC, USA) and IBM SPSS Statistics 23.0.

## 3. Results

### 3.1. Patient Population

The GKRS database included 111 patients with solitary CCMs between February 2005 and March 2020. The median age was 42.6 years old (range = 9.8–79.7) and females comprised 52.2% (male = 54; female = 57) of the cohort. The mean age in the seizure group was younger than in the nonseizure group (32.1 vs. 43.9 years old; *p* = 0.01). The symptoms at presentation were as follows: seizure (*n* = 11), headache/dizziness (*n* = 31), paresis or paresthesia (*n* = 44), other symptoms including cranial nerves symptom, aphasia, and hydrocephalus (*n* = 22), and incidental (*n* = 3). In the seizure group, 11 of the 12 patients included seizure at the initial presentation, and the remaining patient in the seizure group initially presented hemiparesis and developed seizures 5 years later in the postpartum setting. The locations of the 111 CCMs were classified via computerized image processing: 17 lesions were found in the frontal lobe (15%), 2 in the temporal lobe (2.7%), 4 in the parietal lobe (3.5%), 2 in the occipital lobe (1.8%), 18 in the sublobar region (15.9%), 3 in the limbic lobe (2.7%), 11 in the cerebellum (9.7%), and 54 in the brainstem (48.7%). All CCMs in the seizure group were supratentorial. (Table 1). Figure 2 illustrates the volume, proportion, and location of CCMs in two patients: Figure 2A illustrates a left mesial temporal CCM with epilepsy and hemorrhage, and Figure 2B illustrates a left basal ganglia CCM without epilepsy and with hemorrhage.

### 3.2. Volume and Proportion of GM Displaced by the CCM

T1WI images were normalized to the MNI space to eliminate individual differences in brain size and shape. In the seizure group, the mean volume of displaced GM was 3.103 mL, and the mean proportion of displaced GM was 49.7%. In the nonseizure group, the mean volume of displaced GM was 0.613 mL, and the mean proportion of displaced GM was 16.3%. Significant differences were observed between the two groups in terms of the volume and proportion of displaced GM (*p* < 0.00001 and *p* < 0.00001, respectively; Table 2; Figure 3A). Figure 3B,C illustrate the relationship between the proportion or volume of displaced GM and the incidence of seizure. The risk of seizure was higher in cases where the proportion of GM displaced by the CCM exceeded 31% (sensitivity 0.917, 95% CI 0.905–0.929; specificity 0.838, 95% CI 0.816–0.860; PPV 0.407, 95% CI 0.368–0.446; NPV 0.988, 95% CI 0.986–0.990; F1-score 0.564, 95% CI 0.524–0.604; DOR 0.570, 95% CI 0.530–0.610; MCC 0.546, 95% CI 0.506–0.586; kappa coefficient 0.487, 95% CI 0.446–0.528; AUC 0.901, 95% CI 0.823–0.978) (Figure 4 and Appendix A). The calibration slope of calibration plot was 0.892 (Appendix A). The Hosmer–Lemeshow test was also used to assess the goodness of fit pertaining to logistic regression models. The relevant R^2^ value was 0.412 and chi-square was 6.324 with 8 degrees of freedom and a *p* value of 0.611. Gross et al. [10] reported that 37% of CCM patients presented with seizures. The predicted PPV of the model in real practical situations was 0.769. The risk of seizures was also strongly correlated with the volume of displaced GM. Sensitivity analysis excluding patients with brainstem CCMs revealed a similar trend in which the risk of seizure was proportional to the volume of GM.

### 3.3. Percentage of CCM Located within the Standard Brain Space

After normalizing the images to the MNI space, T1WI images were fitted to the atlas to determine the coverage of the CCMs. We then analyzed the location coverage in the seizure and nonseizure groups (Appendix A). In the seizure group, a statistically significant proportion of the CCMs were located over the frontal, temporal, parietal, occipital, and limbic lobes (*p* < 0.05). In the nonseizure group, most of the CCMs were located over the brainstem (*p* < 0.001) or cerebellum (*p* = 0.062). The seizure and nonseizure groups were similar in terms of sublobar distributions.

### 3.4. Subgroup Analysis of CCM Location and Relative Seizure Risk

We also classified the patients into supratentorial and infratentorial groups based on CCM location (Table 3). The supratentorial group comprised patients with CCMs adjacent to the frontal lobe, temporal lobe, parietal lobe, occipital lobe, limbic lobe, or sub-lobar region. The infratentorial group comprised patients with CCMs adjacent to the cerebellum and brainstem. Only the patients with supratentorial CCMs experienced seizures. We also categorized the supratentorial group into temporal and extra-temporal groups. The temporal group included patients with CCMs adjacent to the temporal or limbic lobes. The extra-temporal group included patients with CCMs outside the temporal lobes. The risk of seizure was 5.3 times higher in the temporal group than in the extratemporal group (OR = 5.3, *p* = 0.09).

## 4. Discussion

### 4.1. Volume and Proportion of Displaced GM as Predictors of Epileptogenicity

Cortical involvement in CCMs is a well-known risk factor for seizure presentation and development; however, a correlation between the volume or proportion of cortical involvement and the relative risk of seizure has not been proven. In this study, we developed a computer algorithm capable of quantifying the volume and proportion of GM displaced by CCMs [7]. The volume of displaced GM was strongly correlated with the risk of seizure in cases where the proportion of displaced GM exceeded 31%. In sensitivity analysis, the exclusion of brainstem CCMs had no effect on this trend. This insight could help in estimating the risk of seizure risk in CCM patients. Note that CCM patients with a very low volume of displaced GM were less likely to present/develop seizures, regardless of the proportional displacement (Figure 3A). This suggests a minimum threshold of displaced volume for seizure presentation/development.

There has been some debate as to the clinical necessity of antiepileptic drugs for incidental or nonseizure presenting CCM. Josephson et al. [11] reported that the 5-year risk of seizure after presentation with incidental CCM was 4%. They also reported that the 5-year risk of seizure after presentation with intracranial hemorrhage or focal neurologic deficits was 6%. This corresponds to our results in which only 1 of the 12 epilepsy patients did not exhibit seizures at the initial presentation. In cases of CCM, seizure prophylaxis is generally unnecessary; however, the volume and proportion of displaced GM may provide guidance in managing seizure risk. Our results indicate that a CCM displacing less than 31% of the GM presents a very low risk of seizure risks (NPV = 0.988), such that prophylactic anticonvulsant is not required. Note that we also created a graphical user interface (GUI) to facilitate clinical integration (Appendix A).

### 4.2. CCM Location and Epileptogenicity

The volume and proportion of displaced GM are not the only factors affecting the incidence of seizure in cases of CCM. The structural and functional impact of the CCM within brain networks can also play an important role in focal epileptogenicity. In the current study, all of the CCMs in the seizure group were supratentorial, and the sub-lobar location of the CCMs was associated with a lower risk of seizure. Note however that the risk of seizure was higher in cases where the CCM was adjacent to the limbic lobe. In subgroup analysis, temporal lobe involvement (including temporal and limbic lobes) was related to an elevated risk of seizure. Note also that previous studies reported that seizures occurred more frequently in cases where the CCM involves the cortex [3]. Menzler et al. [2] reported that the epileptogenicity of CCMs is associated with mesiotemporal archicortical involvement, which is consistent with our findings.

### 4.3. Potential Discrepancies between CCM Images and Gray Matter Density Template

The proposed computer-assisted algorithm uses 3D FSPGR images for clinical diagnosis; however, obtaining accurate estimates pertaining to the volume and proportion of the GM distribution in 3D FSPGR or T1WI images can be difficult. It is for this reason that conventional brain tissue segmentation algorithms often fail in their analysis of GM [12,13,14,15].

Our primary objective in this study was to estimate the volume and proportion of GM using a standard GM density template with CCM image labels subjected to spatial normalization. Our normal GM density template was collected from 22 normal control subjects, with a mean age of 25.3 (20–30) years old and a standard deviation of 4.4 years. The mean age of our study subjects was 32.1 (9.8–52.9) years old in the seizure group and 43.9 (14.26–79.7) years old in the nonseizure group. Note the difference in age between CCM patients and the subjects of the standard GM template. Aging has been shown to affect global brain volume, including decreases in hippocampal, temporal, and frontal lobes and increases in the ventricle and CSF spaces [16]. Scahill et al. [16] conducted a longitudinal study of changes in brain volume during the normal aging process. Their results revealed whole brain atrophy of roughly 0.32% per year, with atrophy of 0.68% and 0.82% per year, respectively, in the temporal lobes and hippocampus. Note that the age-related loss of GM is disproportionate to that of WM. Benedetti et al. [17] reported an average loss of 2.8 mL of gray matter per year and a 1.1 mL loss of WM. The young population in the current study, however, should have minimized the effects of age on estimates of GM volume.

### 4.4. Limitations

The major limitation of this study was the small sample size. Note however that despite the small sample size, we observed a clear correlation between displacement volume and the risk of seizures. In addition, the fact that all of the CCM cases were collected from a GKRS database compiled by our institution may have resulted in selection bias. Essentially, CCM patients with lesions that were less amenable to surgery and those with milder symptoms were selected to undergo GKRS, and patients with multiple CCMs were excluded in order to isolate the relationship between CCM location and seizure risk.

## 5. Conclusions

The cortical involvement of CCMs in seizure incidence can be quantified using the volume and proportion of GM. A higher volume or proportion of GM was associated with an elevated risk of seizure presentation/development. Determining the risk of seizure using volume or proportion of displaced GM could guide the selection of seizure prophylaxis for CCM patients.

## Figures and Tables

**Figure 1 biomedicines-09-01872-f001:**
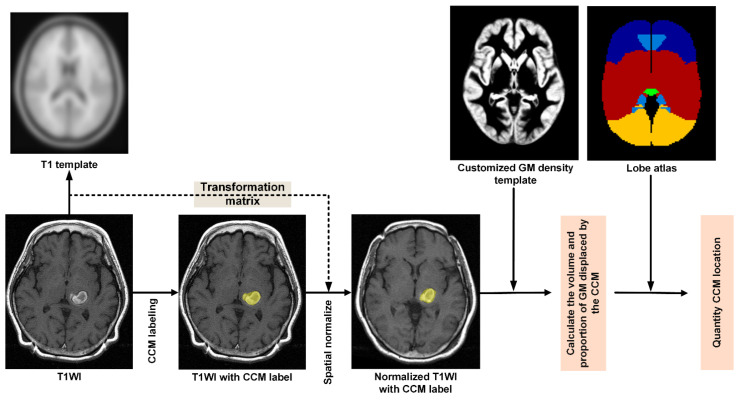
Flowchart showing process of image processing and CCM qualification: Manual delineation of CCM, spatial normalization, construction of customized GM density template, quantifying the displacement of GM by CCM, and estimating the relative location of CCM.

**Figure 2 biomedicines-09-01872-f002:**
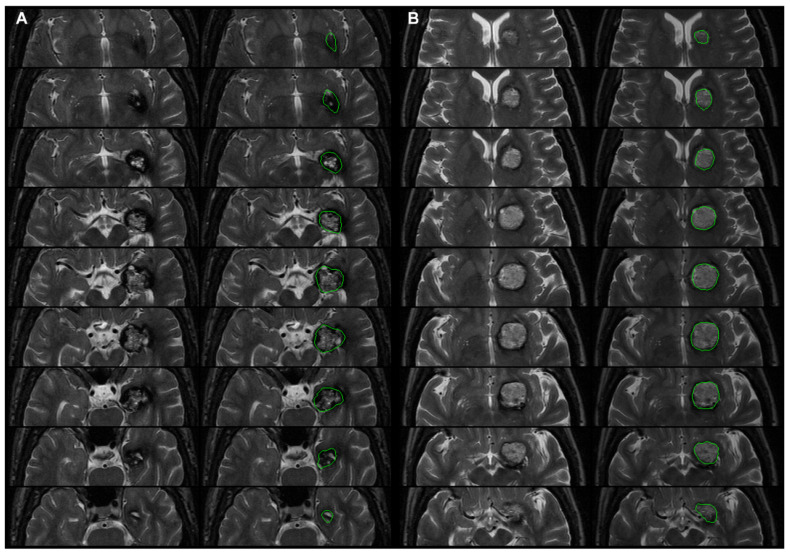
Nine successive axial slices obtained from the CCM of two patients: (**A**) left mesial temporal CCM with epilepsy and hemorrhage, and (**B**) left basal ganglia CCM without epilepsy and with hemorrhage (left column: zoomed in axial T2WI; right column: green contours being the CCM demarcated by an experienced neurosurgeon). The volume and proportion of GM displaced by the CCM were as follows: (**A**) 4.92 mL and 65.2%, and (**B**) 3.57 mL and 33.97% in MNI space. (**A**) CCM location in limbic lobe (58%), temporal lobe (12%), and sublobar (29%); (**B**) CCM location in frontal lobe (6%), limbic lobe (4%), and sublobar (89%) in MNI space.

**Figure 3 biomedicines-09-01872-f003:**
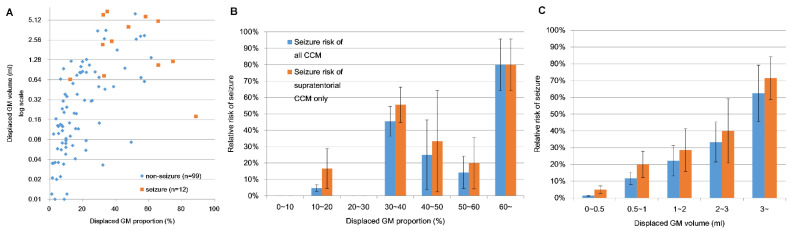
(**A**) Clinical correlations between proportion of displaced GM and volume of displaced GM; (**B**) Relationship between relative risk of seizure and proportion of displaced GM; (**C**) Correlation between relative risk of seizure and volume of displaced GM in MNI space. The 95% confidence interval was revealed by the sample mean ± 1.96 times its standard error.

**Figure 4 biomedicines-09-01872-f004:**
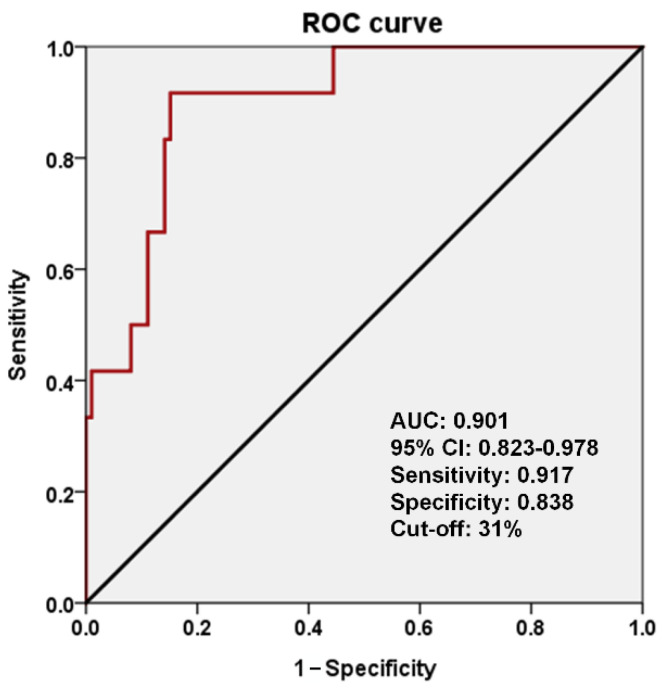
The risk of seizure was higher in cases where the proportion of GM displaced by the CCM exceeded 31% (sensitivity 0.917, 95% CI 0.905–0.929; specificity 0.838, 95% CI 0.816–0.860, AUC 0.901, 95% CI 0.823–0.978). Receiver operating characteristic (ROC), area under the curve (AUC), and confidence intervals (CI).

**Table 1 biomedicines-09-01872-t001:** Characteristics of seizure and nonseizure patients.

Characteristics	Total (*n* = 111)	Seizure (*n* = 12)	Nonseizure (*n* = 99)	*p*	OR (95%CI)
Age (range)	42.6 (9.8–79.7)	32.1 (9.8–52.9)	43.9 (14.26–79.7)	0.01	0.94 (0.89–0.99)
Gender	M:54 F:57	M:6 F:6	M:48 F:51	0.92	1.06 (0.32–3.52)
Volume of CCMs (mL) (with hemosiderin)	2.79	6.09	2.39	0.001	1.27 (1.09–1.47)
Initial symptoms					
Seizure	11	11	0	<0.001	∞ (317.83–∞)
Headache/dizziness	31	0	31	0.19	0 (0–0.73)
Paresis/paresthesia	44	1	43	0.04	0.12 (0.02–0.95)
Others	22	0	22	0.12	0 (0–1.17)
Incidental finding	3	0	3	1	0 (0–11.28)
Location of CCMs					
Frontal lobe	17	5	12	0.01	5.18 (1.42–18.94)
Temporal lobe	2	1	1	0.13	8.9 (0.52–152.60)
Parietal lobe	4	2	2	0.03	9.7 (1.23–76.50)
Occipital lobe	2	1	1	0.13	8.9 (0.52–152.60)
Sublobar	18	1	17	0.44	0.44 (0.05–3.63)
Limbic lobe	3	2	1	0.02	19.6 (1.63–235.70)
Cerebellum	11	0	11	0.61	0 (0–2.73)
Brainstem	54	0	54	<0.001	0 (0–0.28)

Note: ∞ = infinity; M = Male; F = Female.

**Table 2 biomedicines-09-01872-t002:** Volume and proportion of GM displaced by CCM in MNI space.

Gray Matter	Seizure (*n* = 12)	Nonseizure (*n* = 99)	*p*
Volume (mL)	3.024 ± 2.415	0.521 ± 0.970	<0.00001 *
Proportion (%)	48.540 ± 22.076	15.938 ± 15.59	<0.00001 *

* *p* value <0.05 indicated a statistical significance via the Wilcoxon rank sum test.

**Table 3 biomedicines-09-01872-t003:** Subgroup analysis of CCM location and corresponding risk of seizure.

Location Group	Seizure	Nonseizure	OR (95%CI)	*p*
* Total (*n* = 111)	(*n* = 12)	(*n* = 99)		
Infratentorial (*n* = 65)	0	65	1	-
Supratentorial (*n* = 46)	12	34	-	-
** Supratentorial (*n* = 46)	(*n* = 12)	(*n* = 34)		
Extratemporal (*n* = 41)	9	32	1	-
Temporal (*n* = 5)	3	2	5.3 (0.77–36.96)	0.09

* Supratentorial = frontal + temporal + parietal + occipital + sublobar + limbic lobe; Infratentorial = cerebellum + brainstem. ** Extratemporal = frontal + parietal + occipital + sublobar; Temporal = temporal + limbic lobe.

## Data Availability

Not applicable.

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
