# Peer review of "Displacement of Gray Matter and Incidence of Seizures in Patients with Cerebral Cavernous Malformations"

_biomedicines, 2021, doi:10.3390/biomedicines9121872_

Round 1

Reviewer 1 Report

It is an interesting paper in which the seizure risk is determined based on the volume and/or proportion of gray matter (GM). The authors used logistic regression and provided the ORs of the factors. The reviewer provided some critical issues, needed to be addressed and implemented by the authors. Otherwise, the paper cannot be suitable from the clinical and prediction point of view:

-No discrimination and calibration

-No proper validation

Based on a variety of papers, it is necessary to report discrimination and calibration of the (risk assessment/prediction) method:

https://www.equator-network.org/reporting-guidelines/tripod-statement/

https://bmcmedicine.biomedcentral.com/articles/10.1186/s12916-019-1466-7

https://link.springer.com/book/10.1007/978-0-387-77244-8

  1. The discrimination between seizure and non-seizure groups, based on the probability values obtained using the logistic regression: The ROC curve, the AUC value including the CI 95%
  2. A proper cut-off must be provided for the probability values, and the following indices with their CI 95% must be reported:

Sensitivity, Specificity, PPV, NPV, F1-score, DOR, MCC, and Kappa

  1. A calibration plot must be shown, in which the calibration slope (and calibration in large) must be reported.
  2. The goodness-of-fit of the logistic regression (relevant R2 and the Hosmer–Lemeshow test) must be provided.
  3. Based on the prevalence of the seizure in the population, the Bayes’ rule must be used to estimate the method’s performance in a practical situation, as the dataset does not resemble the true prevalence of the disease. It could be done as the following:

P(D\E)=(Se*P)/(Se*P+(1-SP)*(1-P));

D: The person has a seizure

E: The proposed system predicted the seizure

P=true prevalence of the seizure in the population

Se and Sp are obtained sensitivity and specificity on the samples

In fact, P(D\E) is the predicted PPV of the system in practice.

Note: The most important quantitative indices must be added to the abstract.

Reviewer 2 Report

Seizures are the most common presenting symptom of cerebral cavernous malformations (CCMs) and usually progress to medically refractory epilepsy. The authors propose an algorithm to estimate the volume and proportion of the displaced gray matter (GM) in order to predict seizure incidence of CCMs. The approach is based on the hypothesis that the displaced volume of GM by CCM is associated with risk of seizure.

The article is well structured and the presentation of the study is adequately described.

The algorithm steps are clearly presented. A limitation of the proposed algorithm is that the CCM contour is not automatically detected but manual intervention is needed to obtain it.

The results show a correlation between the displacement volume and the risk of seizures. However, my main concern is related to the fact that the number of subjects included in the study is limited, and there is not enough data to reach statistical conclusion and to validate the hypothesis introduced by the authors. Thus, for publication in high impact factor journal a larger sample size should be used and patients from different medical centers should be included.

Minor comments:

  1. For clarity and readability, the formulas introduced in page 3 lines146-149 should be expressed as equations separated from the text paragraph.
  2. Page 8, line 227 the (Figure 4, Video 1) should be removed.

Reviewer 3 Report

In this study, based on the hypothesis that the volume and percentage of gray matter (GM) displaced by cerebral cavernous malformations (CCM) is associated with the risk of seizure, Chou et al. developed an algorithm to quantify the volume and percentage of displaced GM and the risk of seizure. Image analysis of 111 patients with solitary CCMs (divided into seizure and non-seizure groups) enrolled in the Gamma Knife radiosurgery database at the authors' hospital showed that the CCM algorithm was effective in quantifying the GM and CCM using T1WI MR images. In the seizure group, 11 of 12 patients exhibited seizure at the initial presentation, and all CCMs in the seizure group were supratentorial. The location of the limbic lobe within the CCM was significantly associated with the risk of seizures (OR = 19.6, p = 0.02). The risk of seizures increased when the percentage of GM displaced by the CCM exceeded 30%. It was also strongly correlated with the volume of the displaced GM. Both the volume and percentage of displaced GM were positively correlated with the risk of seizure presentation/development and could potentially guide seizure prophylaxis in CCM patients.

   Basically, the authors have been working on an important study, and the results are considered reasonable. However, some problems were observed in the current manuscript as shown below, and I recommend that the paper should be revised before it is published in Biomedicines.

  1. In recent years, domain shift, which causes discrepancies among institutions in computer image analysis, has become a problem (Pooch et al., 2020, arXiv:1909.01940: https://arxiv.org/pdf/1909.01940.pdf). Although all the results of this study were conducted using only images obtained at the authors' hospital, the robustness of the study is unknown. For example, can MR images obtained at other facilities show similar results to those in this paper?
  2. It is better to mention the medical considerations regarding the higher volume and/or percentage of GM, the higher the risk of seizure presentation/development.
  3. In this study, the authors have used MATLAB and SPM12 to normalize T1WI images and CCM image labels into MNI space, but I wonder if the latest deep learning techniques can be used to create a more advanced platform.

Author Response

Response to Reviewer 3 Comments

In this study, based on the hypothesis that the volume and percentage of gray matter (GM) displaced by cerebral cavernous malformations (CCM) is associated with the risk of seizure, Chou et al. developed an algorithm to quantify the volume and percentage of displaced GM and the risk of seizure. Image analysis of 111 patients with solitary CCMs (divided into seizure and non-seizure groups) enrolled in the Gamma Knife radiosurgery database at the authors' hospital showed that the CCM algorithm was effective in quantifying the GM and CCM using T1WI MR images. In the seizure group, 11 of 12 patients exhibited seizure at the initial presentation, and all CCMs in the seizure group were supratentorial. The location of the limbic lobe within the CCM was significantly associated with the risk of seizures (OR = 19.6, p = 0.02). The risk of seizures increased when the percentage of GM displaced by the CCM exceeded 30%. It was also strongly correlated with the volume of the displaced GM. Both the volume and percentage of displaced GM were positively correlated with the risk of seizure presentation/development and could potentially guide seizure prophylaxis in CCM patients.

Basically, the authors have been working on an important study, and the results are considered reasonable. However, some problems were observed in the current manuscript as shown below, and I recommend that the paper should be revised before it is published in Biomedicines.

Point 1: In recent years, domain shift, which causes discrepancies among institutions in computer image analysis, has become a problem (Pooch et al., 2020, arXiv:1909.01940: https://arxiv.org/pdf/1909.01940.pdf). Although all the results of this study were conducted using only images obtained at the authors' hospital, the robustness of the study is unknown. For example, can MR images obtained at other facilities show similar results to those in this paper?

Response 1: We believe that the processing of images (Methods 2.3, Stage 2 - Preprocessing of T1WI images from patients), including normalizing to the MNI space and quantifying the displacement of GM can be performed using different MRI via a single numerical calculation. Nonetheless, the preparation of control images (Method 2.3, Stage 1 - Preprocessing control MPRAGE images) is likely to differ somewhat among MRI machine models. It may be better to create a dedicated GM template for every MRI model and then perform age matching of the patient population. We also hope that it will be possible to perform validation in multiple centers in the near future.

Point 2: It is better to mention the medical considerations regarding the higher volume and/or percentage of GM, the higher the risk of seizure presentation/development.

Response 2: Indeed, the final goal of our study was to use our results to guide clinical practice. For example, a larger-sized CCM with higher risk of seizures would imply the need for surgical intervention. Smaller CCMs with a lower risk of seizures could simply be monitored or subjected to radiation. Note however that in our case series from a single intervention subgroup (e.g. gamma knife case series), the CCMs were relatively small and seizure risk was generally low. For patients with large volume CCMs, the incidence of seizure is generally higher. We propose that cases where the CCM displaces less than 31% of the GM do not require a prophylactic anticonvulsant. Nonetheless, we were unable to obtain a conclusive indication of the treatment modality required for CCMs with a high risk of seizure. We hope that it will be possible to obtain statistically meaningful results from a large patient population in the near future.

Point 3: In this study, the authors have used MATLAB and SPM12 to normalize T1WI images and CCM image labels into MNI space, but I wonder if the latest deep learning techniques can be used to create a more advanced platform.

Response 3: In future work, we will present a three-dimensional Convolutional Neural Network for the challenging task of CCM lesion segmentation. In this study, we have developed an algorithm by which to automatically quantify the volume and proportion of displaced GM and the risk of seizure. We will combine the two algorithms to create a more advanced platform.

Round 2

Reviewer 1 Report

The authors revised the paper based on the reviewer's comments. However, it is not clear why the calibration plot in the x and y-axis is not limited to [0,1] as the observed and predicted risks. This issue must be considered and the plot must be corrected. Also, The CI of 95% lines must be added to the plot.

The required functions in R "val.prob.ci.2" could be assessed from:

https://link.springer.com/book/10.1007/978-0-387-77244-8

Reviewer 3 Report

Basically, I feel that the authors have effectively addressed the criticisms raised in the initial review. I recommend publication.

Author Response

Thank you very much.